# COUP-TFII in Kidneys, from Embryos to Sick Adults

**DOI:** 10.3390/diagnostics12051181

**Published:** 2022-05-09

**Authors:** Sumiyasu Ishii, Noriyuki Koibuchi

**Affiliations:** Department of Integrative Physiology, Gunma University Graduate School of Medicine, Maebashi 371-8501, Japan; nkoibuch@gunma-u.ac.jp

**Keywords:** chicken ovalbumin upstream promoter–transcription factor (COUP-TF), orphan nuclear hormone receptor, kidney, organogenesis, acute kidney injury, podocyte

## Abstract

Chicken ovalbumin upstream promoter-transcription factor II (COUP-TFII) is an orphan nuclear hormone receptor of unknown ligands. This molecule has two interesting features: (1) it is a developmental gene, and (2) it is a potential hormone receptor. Here, we describe the possible roles of COUP-TFII in the organogenesis of the kidneys and protection from adult renal diseases, primarily in mouse models. COUP-TFII is highly expressed in embryos, including primordial kidneys, and is essential for the formation of metanephric mesenchyme and the survival of renal precursor cells. Although the expression levels of COUP-TFII are low and its functions are unknown in healthy adults, it serves as a reno-protectant molecule against acute kidney injury. These are good examples of how developmental genes exhibit novel functions in the etiology of adult diseases. We also discuss the ongoing research on the roles of COUP-TFII in podocyte development and diabetic kidney disease. In addition, the identification of potential ligands suggests that COUP-TFII might be a novel therapeutic target for renal diseases in the future.

## 1. Introduction

Hormones are produced by the endocrine organs and released into the blood. Although the entire body is exposed to hormones delivered by bloodstream, only the target cells express receptors specific to each hormone and mediate the actions of hormones. Hormones are divided into two groups according to their chemical nature: hydrophilic and lipophilic hormones. Hydrophilic hormones include peptide hormones and some amino acid–derived hormones, such as catecholamine. These hormones bind to receptors located on the cell membrane and initiate signal transduction. The receptors for lipophilic hormones are localized in the cytosol and nuclei. On binding of the ligand, the cytosolic receptors enter the nuclei and regulate the transcription of the target genes. In general, the hormone receptors localized in the nuclei suppress the transcription of the target gene in the absence of the ligand and stimulate transcription on ligand binding. These cytosolic and nuclear receptors, which serve as ligand-dependent transcription factors, are called nuclear hormone receptors [1].

Nuclear hormone receptors bind to the specific DNA sequence on the target genes and stimulate or suppress transcription. The structures of these receptors are similar to each other [2,3], although there are at least 65 members in the nuclear hormone receptor superfamily [4,5]. The N-terminal region exhibits a function of ligand-independent autonomous transcriptional activation [6]. The central part is the DNA-binding domain. This domain contains two zinc finger motifs and typically binds to the hexameric DNA motif PuGGTCA (Pu means A or G) [7,8]. The ligand binding domain is located at the C-terminus of the receptors. This domain changes the structural conformation on ligand binding [9,10,11]. The unliganded structure favors the interaction with transcriptional corepressors, and the liganded receptors are suitable for binding to coactivators. These coregulator exchanges are responsible for ligand-dependent transcriptional regulation [12,13,14].

In addition to these ligand-regulated receptors, there are many orphan nuclear hormone receptors, which conserve the nuclear hormone receptor structure. However, their ligands are yet to be identified [15]. These molecules work as transcription factors without ligand dependency. Although some receptors are transcriptional activators, others work as transcriptional repressors. Their functions are dependent, at least in part, on which molecules they interact with: coactivators or corepressors. As the ligand specificity of orphan nuclear hormone receptors is high, when natural or synthetic ligands are identified, they may be regarded as potential therapeutic targets of multiple diseases. Chicken ovalbumin upstream promoter–transcription factors (COUP-TFs) are orphan nuclear hormone receptors of unknown ligands.

There are two members in mammalian COUP-TFs: COUP-TFI (NR2F1) and COUP-TFII (NR2F2) [16]. In this review article, we describe the possible roles of COUP-TFII in the organogenesis of kidneys and protection from renal diseases primarily in mouse models. First, we provide an overview of the general aspects of COUP-TFs. Next, we explain the role of COUP-TFII in embryonic kidney development. Then, we describe the role of COUP-TFII in protecting from acute kidney injury (AKI). Finally, we provide some ongoing stories regarding podocytes and diabetic kidney disease.

Spatiotemporally specific transcriptional regulation plays a pivotal role in organogenesis. In many organs, COUP-TFs are regarded as developmental genes. Researchers have postulated that the role of many developmental genes is limited in adults because of the dramatic decrease in expression levels. However, developmental genes exhibit new functions in the etiology of adult diseases [17]. In this regard, we explain in this article the importance of COUP-TFII in the kidneys from embryos to sick adults.

The kidneys are well known as an endocrine organ producing renin and erythropoietin. Kidneys are also an important target organ of mineralocorticoids. The understanding of COUP-TFII as an orphan hormone receptor might provide some additional suggestions about the interaction between the kidneys and the endocrine system.

## 2. COUP-TFs

### 2.1. Overview

COUP-TFI was identified as a member of the nuclear hormone receptor superfamily [18]. It was independently cloned as a gene related to oncogene v-erbA [19]. This molecule was found to bind chicken ovalbumin upstream promoter and stimulate transcription together with another protein; therefore, it is called COUP-TF [20]. COUP-TFII was subsequently identified as a gene related to COUP-TFI [21]. It was also independently cloned as a regulator of apolipoprotein A1 gene transcription [22]. The amino acid sequences are highly conserved between COUP-TFI and COUP-TFII. The sequence homology is 98% in the DNA-binding domain and 96% in the putative ligand binding domain [16,23]. These similarities are evident compared with other nuclear hormone receptors.

Typically, nuclear hormone receptors work as homodimers or heterodimers [24]. COUP-TFs favor homodimer formation in solution and on DNA [22,25], although researchers have reported heterodimerization with other nuclear hormone receptors, such as retinoid X receptor [26]. The most common binding sequence of COUP-TFs is the direct repeat of two PuGGTCA sequences with one nucleotide spacer, which is called direct repeat 1 [27]. COUP-TFs interact with nuclear receptor corepressors containing histone deacetylase 3 rather than coactivators and suppress the transcription of target genes [14,28]. On the other hand, a certain number of genes are upregulated by COUP-TFs. One interesting feature is that COUP-TFs interact with another transcription factor, Sp1, and stimulate the expression of Sp1-target genes [29].

COUP-TFs are highly expressed in mouse embryos. They are detectable on embryonic day (E) 7.5. Their expression levels then increase to reach a peak on E14–15 and then dramatically decrease after birth [30]. Although the expression patterns of COUP-TFI and COUP-TFII overlap, they are still different [16]. The expression levels of COUP-TFs are much lower in adult mice than in embryos [31].

### 2.2. COUP-TFI

COUP-TFI is dominantly expressed in the developing central nervous system of mouse embryos. Consistent with these findings, *COUP-TFI* knockout mice experience perinatal death and show multiple neural defects, including defects in the morphogenesis of the glossopharyngeal ganglion, differentiation of subplate neurons and guidance of thalamocortical axons, early regionalization of the neocortex, and formation of commissural projections in the forebrain [32,33,34,35].

In humans, heterozygous mutations in the *COUP-TFI* gene were identified in patients with Bosch–Boonstra–Schaaf optic atrophy syndrome, characterized by global developmental delay, moderately impaired intellectual development, and optic atrophy [36,37]. Concordantly, defective optic vesicle morphogenesis was caused by the eye-specific deletion of both the *COUP-TFI* and *COUP-TFII* genes in the mouse [38]. This phenotype was not observed in mice deficient in each single gene, which suggests compensatory effects between the two COUP-TFs.

### 2.3. COUP-TFII

COUP-TFII is broadly expressed in developing murine embryos, particularly in the stroma cells. In accordance with these findings, COUP-TFII plays important roles in the development of multiple mouse organs [39]. As a result of defects in angiogenesis and cardiogenesis, conventional *COUP-TFII* knockout mice die at about E10 [40]. Therefore, the conditional knockout system, heterozygous null mutants, or a tissue-specific overexpression system have been used to study the functions of COUP-TFII in vivo. COUP-TFII is essential for venous endothelial cell formation [41,42]. It is also important for the development of lymphatic vessels [43]. COUP-TFII is responsible for reproductive function in both males [44] and females [45,46,47] and for the elimination of the male reproductive tract in female embryos [48]. COUP-TFII is involved in adipogenesis and energy metabolism [49,50,51,52]. Studies have reported the roles of COUP-TFII in the development of the diencephalon [53], cerebellum [54], and eyes [38,55]. COUP-TFII is also indispensable for stomach patterning [56] and limb and skeletal muscle development [57,58].

Consistent with the importance of the developing mouse heart, multiple types of variants were identified in the *COUP-TFII* gene in human patients with multiple types of congenital heart defects [59,60,61]. Some of these patients also exhibited congenital diaphragmatic hernia [62]. In addition, one of the most common chromosome anomalies found in patients with congenital diaphragmatic hernia is deletion in human chromosome 15q26, where the *COUP-TFII* gene resides [63,64]. Interestingly, deletion of the *COUP-TFII* gene in the mouse foregut mesenchyme region results in Bochdalek-type congenital diaphragmatic hernia [65,66].

In addition to the important roles of COUP-TFII in developing embryos, animal studies have shown COUP-TFII’s implication in several diseases, including obesity, cancer, muscular dystrophy, and endometriosis [49,67,68,69]. These findings suggest that developmental genes, such as *COUP-TFII*, exhibit new functions in the etiology of adult diseases.

Among multiple tissues, COUP-TFII is highly expressed in the kidney [70,71]. These findings strongly suggest that COUP-TFII plays an important role in the kidney.

## 3. COUP-TFII in Kidney Organogenesis

### 3.1. Kidney Development

Although the kidneys have a complicated structure and function, their developmental procedure has been well studied both in both humans and mice [72,73,74,75]. Mammalian kidneys are derived from the intermediate mesoderm and organized through three stages: pronephros, mesonephros, and metanephros [76]. Pronephros is a transient structure that appears at about 3 weeks of gestation in humans and at E8.5–9 in mice. The mesonephros emerges as the second transient kidney at 3–4 weeks of gestation in humans and at E9 in mice. The metanephros is the primordium of the permanent kidney that is formed at 5 weeks’ gestation in humans and at E10.5 in mice.

The metanephros consists of two types of structures: the ureteric bud and metanephric mesenchyme, both of which are derived from the intermediate mesoderm. Some of the intermediate mesoderm cells undergo a mesenchymal–epithelial transition and differentiate into the epithelial Wolffian duct. The ureteric bud is a diverticulum of the Wolffian duct that grows into the adjacent metanephric mesenchyme. The interaction between the ureteric bud and metanephric mesenchyme plays a pivotal role in the subsequent nephrogenesis. The metanephric mesenchyme secretes glial cell line–derived neurotrophic factor (GDNF), which stimulates the invasion of the ureteric bud into the mesenchyme. The invaded ureteric bud induces condensation of the adjacent mesenchymal cells, and the condensed cells subsequently become cap mesenchyme cells, which in turn induce further outgrowth and branching of the ureteric bud, resulting in the formation of the collecting duct network. Via mesenchymal–epithelial transition, the cap mesenchyme cells then differentiate into epithelial cells of the nephrons. The primordium of the nephron is called the renal vesicle. The vesicle elongates and forms the comma-shaped body, which subsequently becomes the S-shaped body. The S-shaped body further differentiates into multiple sections of a mature nephron. The distal part of the S-shaped body develops into the distal tubule, the middle part gives rise to the proximal tubule, and the proximal part becomes the glomerulus.

Alongside GDNF, tremendous numbers of growth factors and transcription factors are involved in kidney organogenesis [74,77]. In particular, in the metanephric mesenchyme where COUP-TFII is highly expressed (described in the next subsection), a transcriptional network that involves odd-skipped related transcription factor 1 (OSR1), EYA transcriptional coactivator and phosphatase 1 (EYA1), paired box 2 (PAX2), and SIX homeobox 2 (SIX2) plays an important role in the formation of the metanephric mesenchyme and the expression of the GDNF gene [72]. Another key transcription factor in the development of the kidney is WT1 transcription factor (WT1), which serves as a tumor-suppressor gene in Wilms tumors [75].

### 3.2. Expression of COUP-TFII in the Developing Kidney

The cell type-specific expression of COUP-TFII has been studied extensively in mouse embryonic kidneys [78]. COUP-TFII is detectable in the urogenital ridge of the developing mesonephros as early as E9.5. At E11.5, COUP-TFII is highly expressed in the condensed mesenchyme but not in the ureteric buds. COUP-TFII is also expressed in cells that recently underwent the mesenchymal–epithelial transition, as assessed by co-staining with NCAM, a marker for mesenchyme-derived epithelial cells. The COUP-TFII signal is apparently detectable in the developing nephron, the nephrogenic cortex, and stromal cells on E12.5. Within the primordial nephrons, mesenchyme-derived epithelial cells, including the condensed mesenchyme, the renal vesicle, and the comma-shaped body, are positive for COUP-TFII. On E13.5, the COUP-TFII signal disappears in the middle part of the S-shaped body that becomes the proximal tubule, whereas other parts still express COUP-TFII.

### 3.3. Role of COUP-TFII in Kidney Organogenesis

In developing kidneys, the high levels of COUP-TFII expression strongly suggest its role in kidney organogenesis. To study this hypothesis, a stage-specific deletion of the *COUP-TFII* gene in mouse embryos was performed [79]. *COUP-TFII*-null mutants lack the formation of metanephric mesenchyme. This is consistent with the high levels of COUP-TFII expression in this region. Deletion of the *COUP-TFII* gene at a later stage resulted in decreases in the levels of multiple metanephric mesenchymal molecules, including EYA1 [80], PAX2 [81], SIX2 [82], and WT1 [83]. In addition, greater numbers of apoptotic cells were observed in the metanephric mesenchymal region in these mice. Furthermore, COUP-TFII stimulates the expression of *Eya1* and *Wt1* through an interaction with Sp1 in the rat metanephric mesenchyme cell line. These findings indicate that COUP-TFII is essential for the formation of the metanephric mesenchyme and kidney precursor cell survival.

## 4. COUP-TFII in AKI

### 4.1. Expression of COUP-TFII in Adult Kidneys

In mature kidneys, the nephrons consist of glomeruli, proximal tubules, loops of Henle, and distal tubules. The glomeruli consist of podocytes, mesangial cells, endothelial cells, and epithelial cells of Bowman’s capsule, and they are responsible for filtering the primitive urine. The proximal tubules and the loops of Henle are derived from the middle part of the S-shaped body and are responsible for reabsorption of essential molecules from urine. The distal tubules regulate blood pressure and ion levels. The nephrons are connected to ureteric bud–derived collecting ducts.

Although the expression levels of COUP-TFII decrease dramatically after birth, previous studies have investigated the cell type-specific expression pattern in adult mouse kidney [71,78]. In glomeruli, COUP-TFII is expressed in podocytes, mesangial cells, endothelial cells, and epithelial cells of Bowman’s capsule. In addition, the distal tubules, thick ascending limb (TAL), juxtaglomerular cells, and macula densa cells are positive for COUP-TFII. In contrast, COUP-TFII signals cannot be detected in the proximal tubules and collecting ducts. This expression pattern is consistent with that in mouse embryos. As stated above, the ureteric bud lacks expression of COUP-TFII, and as the development proceeds, the middle part of the S-shaped body loses COUP-TFII signals.

### 4.2. AKI

AKI is a syndrome of a rapid reduction of renal function with multiple etiologies [84,85,86]. It is defined by a ≥0.3 mg/dL increase in serum creatinine (Cr) within 48 h, an increase in serum Cr to ≥1.5 times baseline within the previous 7 days, or oliguria for 6 h [87]. In hospitalized patients, the incidence of AKI ranges from 10 to 15% [88], and the frequencies increase to >50% among patients in the intensive care unit [89]. AKI is associated with high mortality rates of 23% (range, 16–50%) depending on the stage [90,91]. AKI is also associated with high mortality in coronavirus disease 2019 (COVID-19) infection as a severe complication [92]. Although several therapeutic strategies including extracorporeal kidney replacement therapy [93] have been developed, the management of AKI remains challenging.

As AKI comprises a heterogeneous group of conditions with various etiologies, it can be categorized by numerous factors, including underlying disease; reversibility; primary affected compartment of the kidney; pathophysiology; site of disease origin, including prerenal, intrarenal, and postrenal lesion; or mechanism of injury [85]. The injury mechanisms include hypotension, hypoxia, nephrotoxicity, and inflammation. One of the major causes of AKI is tubular injury induced by nephrotoxic drugs. These nephrotoxic agents include aminoglycosides, hydroxyethyl, tenofovir, and cisplatin [94].

Cisplatin is a platinum-based chemotherapeutic agent used widely to treat several cancers in various organs, including the lung, ovary, testis, and bladder. It binds to DNA to induce cross-linking, which activates multiple pathways and culminates in apoptosis [95]. AKI is a major and serious side effect of cisplatin because cisplatin accumulates in tubular cells of the kidneys after excretion into primitive urine [96]. Cellular uptake of cisplatin in the kidneys is mediated by transporters including organic cation transporter 2 and copper transport protein [97,98]. On the other hand, some efflux transporters, such as multi-antimicrobial extrusion protein 1, contribute to the excretion of cisplatin from the renal cells and protection against cisplatin-induced AKI [99]. Cisplatin induces multiple types of cellular damage, including stress by mitochondrial dysfunction and the production of reactive oxygen species [100], dysregulated autophagy [101], immune response, and eventually cell death, including apoptosis [102] and necroptosis [103]. Tumor necrosis factor–alpha (TNF-α) plays a central role in cisplatin-induced AKI [104,105]. TNF-α levels increase in both renal tissues and serum after cisplatin treatment. TNF-α is produced by many types of cells in the kidneys, such as tubular cells, fibroblasts, macrophages, and leukocytes. Zhang et al. reported that TNF-α produced in renal parenchymal cells is responsible for cisplatin-induced AKI [106]. Upregulated TNF-α induces not only inflammatory responses but also the production of reactive oxygen species and the activation of apoptotic pathways, partly through activation of nuclear factor kappa B pathway [107]. High levels of serum TNF-α are associated with high mortality [108]. Preoperative levels of TNF-α are associated with higher risk of AKI, greater lengths of hospital stays, and ventilator use in children who underwent cardiac surgery [109]. The susceptibility of different nephron segments depends on the etiologies of renal injury [85]. The S3 segment of the proximal tubules and TALs are primarily damaged in drug-induced AKI, partly because these are metabolically active sites [110].

### 4.3. Role of COUP-TFII in Cisplatin-Induced AKI

When the *COUP-TFII* gene is systemically deleted in mature mice, no gross abnormality is observed, suggesting that the physiological importance of COUP-TFII in adults is not obvious. These findings might be associated with the fact that the expression levels of COUP-TFII decrease dramatically after birth. However, COUP-TFII might exert some functions in sick conditions, similar to other developmental genes [17]. In addition, the kidney is one of the organs in which the expression levels of COUP-TFII are high, even in adults. Therefore, the roles of COUP-TFII in AKI were studied by introducing a cisplatin-induced AKI model into adult *COUP-TFII* knockout mice [71].

Serum levels of Cr usually increase when the filtration function of the kidneys does not work properly. It is a determinant of the diagnosis and severity staging of AKI [87]. After cisplatin was administered, the increases in serum Cr levels were significantly higher in *COUP-TFII* knockout mice than in control littermates, indicating the reno-protective role of COUP-TFII from cisplatin-induced AKI. Consistent with these findings, *COUP-TFII* knockout mouse kidneys exhibited more severe structural changes, including a thyroid-like appearance, which is induced by the dilation of renal tubules and deposition in the lumen of the tubules (Figure 1). In addition, in a murine model of cisplatin-induced AKI, the kidneys of *COUP-TFII*-null mice contained a greater number of dead cells in both the proximal tubules and TALs. TNF-α levels were also found to be higher in the serum and kidney tissues of knockout animals. Interestingly, the expression level of COUP-TFII decreased in the TALs in the murine model of cisplatin-induced AKI.

Severe cell death in *COUP-TFII* knockout renal tubules might be consistent with the fact that renal precursor metanephric mesenchyme cells exhibit apoptosis in knockout embryos [79]. Mitochondrial dysfunction plays important roles in the etiology of AKI, including the production of reactive oxygen species [100], dysregulation of autophagy [101], and induction of apoptosis [102]. Although the role of COUP-TFII in the regulation of mitochondrial functions is not fully understood, COUP-TFII suppresses mitochondrial oxidative phosphorylation-related gene transcription in hypoxic hepatocellular carcinoma cells [111]. Therefore, it would not be surprising if COUP-TFII protects renal cells against cisplatin-induced cell death by regulating mitochondrial function. In particular, COUP-TFII expressed in the TALs might protect TAL cells from cisplatin-induced damage in a cell-autonomous manner. On the other hand, profound tubular injury was also observed in another susceptible region, the S3 segment, where COUP-TFII is not expressed. Although the precise mechanism of action of COUP-TFII is unclear, one potential hypothesis is that upregulated TNF-α produced in TAL cells that lost COUP-TFII expression might be harmful to other tissues. It is suggested that COUP-TFII directly suppresses TNF-α expression at the transcriptional level, and this is presumably mediated by nuclear receptor corepressors containing histone deacetylase 3 [14,28]. Interestingly, histone deacetylase might be involved in AKI severity [112,113]. In addition, several other nuclear hormone receptors, such as glucocorticoid receptor, peroxisome proliferator-activated receptor-α, peroxisome proliferator-activated receptor-β/δ, estrogen-related receptor-α, vitamin D receptor, estrogen receptor, and mineralocorticoid receptor, are involved in the etiology of AKI [114].

## 5. Potential Roles of COUP-TFII in Kidney

The kidney is a complicated organ in which many types of cells are well organized. As mentioned above, COUP-TFII is expressed in multiple cells with a characteristic pattern, suggesting its cell type-specific functions. Podocytes are an important subset of cells, and they play a pivotal role in the function of glomerular filtration [115]. Podocytes surround the outer layer of the glomerular capillaries and are attached to the glomerular basement membrane. They are highly differentiated epithelial cells with specific branching that consists of long primary processes and peripheral foot processes [116]. The foot processes of neighboring podocytes interdigitate with each other and cover the glomerular capillary surface. The gaps between the foot processes are covered by a membrane-like extracellular structure called the slit diaphragm, which serves as a filtration barrier in the podocyte layer to avoid loss of proteins into urine [117,118]. By producing glomerular basement membrane components, such as collagen IV and laminin, podocytes also contribute to the glomerular barrier. In addition, vascular endothelial growth factor produced in podocytes is essential for the maintenance of the fenestrated endothelial cells at the inner layer of the glomerular capillary walls [119]. Primordial podocytes are columnar epithelial cells. During maturation, the neighboring cells lose their contact, except for the basal part, which interdigitates with each other to form the foot processes [120].

Considering the important role of podocytes, we sought to study the role of COUP-TFII in these cells. Using a conditional knockout system, the *COUP-TFII* gene was deleted specifically in mouse podocytes. As shown in Figure 2, the numbers of the foot processes were reduced in the glomeruli of the *COUP-TFII*–null mice. The morphology of the podocytes appeared immature, suggesting an important role of COUP-TFII in podocyte maturation. More precise analyses are necessary to determine whether the foot processes were effaced or reduced. Higher tendency of urinary albumin-Cr ratio was observed in 8-week-old podocyte-specific COUP-TFII knockout mice, but the difference was not statistically significant compared to control littermates. Therefore, we speculate that these mice would not show overt proteinuria at least in adults but would exhibit a delay in the maturation of podocytes. However, the possibility of foot process effacement should not be excluded because it is not always accompanied with proteinuria [121]. Interestingly, several nuclear hormone receptors, such as glucocorticoid receptor, peroxisome proliferator-activated receptor-γ, vitamin D receptor, estrogen receptor, and mineralocorticoid receptor, are involved in the function of podocytes [122]. In addition, histone deacetylases play important roles in podocytes [123]. Thus, it would not be surprising if the COUP-TFII with histone deacetylase 3 complex is shown to regulate podocyte biology.

In addition to AKI, COUP-TFII might be involved in other renal diseases in adults. Diabetic kidney disease (DKD) is the leading cause of kidney replacement therapy worldwide [124]. At the initial stage, the glomerular filtration rate (GFR) increases as a result of the osmotic diuresis induced by hyperglycemia. The increased filtration in the glomeruli is subsequently associated with microalbuminuria, an indicator of early DKD. Subsequently, the GFR begins to fall, and macroalbuminuria appears with the progression of renal dysfunction. When the renal function further deteriorates, overt proteinuria and elevated serum Cr levels are observed, which eventually culminate in severe renal defect. Multiple factors are involved in the etiology of DKD. Established factors include renal hemodynamic changes, ischemia, oxidative stress, inflammatory processes, and an overactive renin–angiotensin–aldosterone system (RAAS). Recently, novel mechanisms, such as genetic and epigenetic regulation, mitochondria dysfunction, and podocyte autophagy have been reported [125].

We aimed to study the role of COUP-TFII in DKD by crossing *COUP-TFII* adult knockout mice with *db/db* mice, an established mouse model of DKD [126]. As shown in Figure 3, deletion of the *COUP-TFII* gene did not affect the albumin–Cr ratios in the urine in this model. These results indicate the non-involvement of COUP-TFII in DKD. However, we cannot totally exclude the possibility that COUP-TFII is involved in other DKD models. One difficulty in this study is that COUP-TFII can also affect glucose metabolism [49]. Many other nuclear hormone receptors are involved in DKD, including the α, β/δ, and γ forms of proliferator–activated receptors, estrogen-related receptor–α, and estrogen receptor [114].

COUP-TFII stimulates the expression of the renin gene [127]. Renin is produced in the juxtaglomerular cells, where COUP-TFII is expressed [78]. It hydrolyzes angiotensinogen secreted from the liver to angiotensin I. Angiotensin I is further cleaved to produce angiotensin II by the angiotensin-converting enzyme. Angiotensin II not only induces vasoconstriction but also stimulates the adrenal gland cortex to secrete aldosterone [128]. This system regulates the concentration of serum electrolytes, arterial blood pressure, and extracellular volume, and is tightly correlated with renal function. The inhibition of the RAAS is an effective therapeutic strategy for decreasing blood pressure and protecting multiple tissues, including the kidneys [129]. In addition, studies have reported the roles of the RAAS or renin itself in renal physiology, pathophysiology, and organogenesis [130,131,132]. Therefore, it is possible that COUP-TFII exerts its multiple functions through the RAAS.

## 6. Conclusions

COUP-TFII is essential for the development and maturation of the kidneys, similar to its roles in other organs. COUP-TFII expression levels are high in primordial kidneys. Although the expression levels are low and functions are unclear in adults, recent studies have provided a possible new role of COUP-TFII in the etiology of adult diseases. For example, COUP-TFII is involved in AKI severity, which suggests its potential role as a therapeutic target. As COUP-TFII is expressed in multiple cell types in the kidney, it is expected that COUP-TFII has many functions that are as yet unknown.

## 7. Future Perspectives

One distinctive feature of hormone receptors is their high specificity to the ligand. Therefore, they are good therapeutic target molecules using small chemical modulators. COUP-TFII is an orphan nuclear hormone receptor whose ligands are yet to be identified. As they are potentially regulated by ligands, orphan nuclear hormone receptors serve as therapeutic targets of small molecules [15]. Much effort has been made to identify natural or synthetic ligands of COUP-TFII. Retinoic acid was reported to be a ligand of COUP-TFII, but high doses were necessary to regulate the function of COUP-TFII [133]. More recently, a small chemical was reported to be a modulator of COUP-TFII function [134]. These reports might lead to the future development of novel therapeutic strategies of renal diseases by modulating the function of COUP-TFII. In addition, the expression level of COUP-TFII decreased in the TALs in the murine model of cisplatin-induced AKI. These findings suggest that the changes in COUP-TFII expression levels might serve as a marker of AKI severity in the future.

Recent advances in biotechnology, material science and nanotechnology enable us to utilize nanomaterials for the treatment of renal diseases (reviewed in [135]). For example, polyethylene glycol-incorporated Mn^2+^–chelated melanin nanoparticles mitigate AKI. It would be of great use if the reno-protective role of COUP-TFII could be applied to nanoparticle-based treatment strategies. The same group also introduced nanomaterials for chronic kidney disease detection [136]. Nanomaterials combined with fluorescence or magnetic resonance imaging successfully detected tumors. Similarly, it would be desired that the changes in COUP-TFII levels, which might serve as a marker of the severity of AKI, will be detectable from outside of the body using nanomaterials in the future. Nanomaterial-based biosensors are also developed for multiple purposes, including determination of Cr levels [137]. It is worth trying these novel technologies to study the functions of COUP-TFII in the future.

## Figures and Tables

**Figure 1 diagnostics-12-01181-f001:**
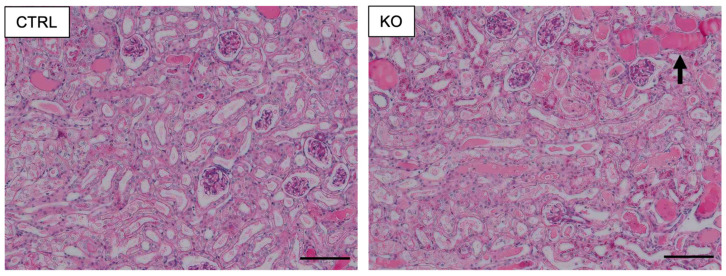
*COUP-TFII*-null animals exhibit more severe morphological abnormalities after cisplatin exposure. CTRL, control; KO, *COUP-TFII* knockout. The thyroid-like appearance in knockouts is indicated by an arrow. Scale bars: 100 μm. Adapted with permission from Ishii et al. [71]. 2020, The Japan Endocrine Society.

**Figure 2 diagnostics-12-01181-f002:**
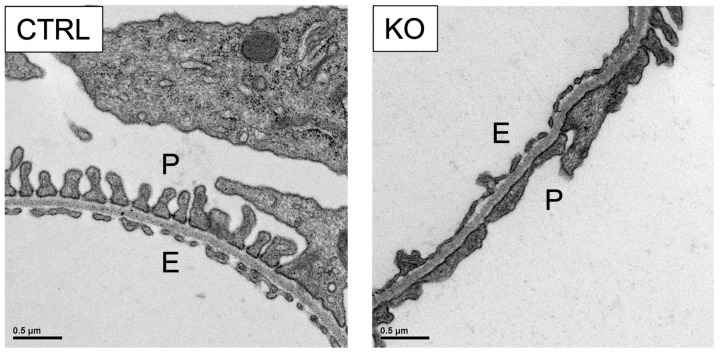
Numbers of foot processes were reduced in the glomeruli of the *COUP-TFII* knockout mice. CTRL control; KO, *COUP-TFII* knockout; P, outer side of the glomeruli where podocytes reside; E, endothelial side. Unpublished data.

**Figure 3 diagnostics-12-01181-f003:**
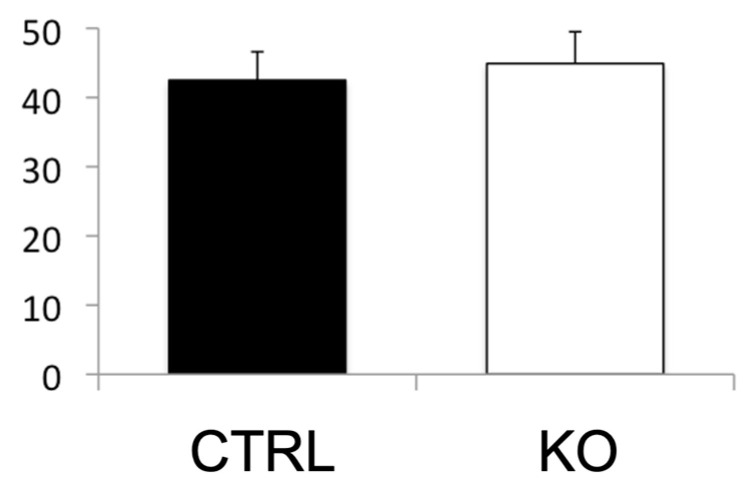
Albumin–creatinine ratios in the urine (µg/mg) did not change in COUP-TFII knockout mice in *db/db* background. CTRL, control; KO, *COUP-TFII* knockout. Unpublished data.

## Data Availability

The data presented in this study are available on request from the corresponding author.

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
