# Peer review of "COUP-TFII in Kidneys, from Embryos to Sick Adults"

_diagnostics, 2022, doi:10.3390/diagnostics12051181_

Round 1
Reviewer 1 Report
1.It is suggested to add new research studies about the role of COUP-TFII in podocyte development and diabetic kidney disease using novel techniques including biosensors.
2.what is the suggestion of this study for future works?
3.Please discuss about the role of organelles including mitochondria in detection of kidney disease using this methods.
4.Please disscuss the possible relation between NfkB pathway and prognosis prediction.
5.Please add these references for your discussion part of manuscript and bold your study novelty :
-Eftekhari, Aziz, et al. "Application of advanced nanomaterials for kidney failure treatment and regeneration." Materials 14.11 (2021): 2939.
-Maleki Dizaj, Solmaz, et al. "Nanomaterials for Chronic Kidney Disease Detection." Applied Sciences 11.20 (2021): 9656.
Author Response
Please refer to an attached file.

Reviewer 2 Report
The authors described the role of not much known COUP-TFII in the kidneys. In Figure 2, the legend says that the number of podocyte foot processes is reduced; are these foot processes effaced or reduced. They seem to be effaced and if effaced, did the mice have nephrotic proteinuria? Does COUP-TFII knockout mice have nephrotic syndrome? Can the authors elaborate on that?
Author Response
Please refer to an attached file.

Round 2
Reviewer 2 Report
The authors have satisfactorily addressed the prior concerns and modified the manuscript.